



# An Optimized and Hybrid Gating Scheme for the Suppression of Very Low Frequency Radios in Transient Electromagnetic Systems

Smith K. Khare[1], Paul McLachlan[2], Pradip Kumar Maurya[3], and Jakob Juul Larsen[1,4]

[1]Department of Electrical and Computer Engineering, Aarhus University, Finlandsgade 22, 8200 Aarhus N, Denmark
[4]WATEC, Aarhus University Centre for Water Technology, Ny Munkegade 120, 8000 Aarhus C, Denmark
[2]Institute for Geoscience, Aarhus University, Denmark
[3]TEM Company, Aarhus, Denmark

**Correspondence:** Jakob Juul Larsen (jjl@ece.au.dk)

**Abstract.** One of the most used approaches for measuring the earth's subsurface resistivity is the transient electromagnetic method (TEM). However, noise and interference from different sources, e.g., radio communication, the instrument, the atmosphere, and powerlines severely taint these types of signals. In particular, radio transmission in the very low-frequency (VLF) range between 3 kHz and 30 kHz is one of the most prominent sources of noise. Transient electromagnetic signals are normally

gated to increase the signal-to-noise ratio. A precise selection of gate shapes is required to suppress undesired noise while allowing the TEM signal to pass unaltered. We employ the multi-objective particle swarm optimization technique to choose optimal gate shapes and placements by minimizing an objective function composed of standard error bars, the covariance between gates, and the distortion of the gated signal. The proposed method is applied to both fully sampled synthetic TEM data and to boxcar-gated field data. The best output from the search space of gate shapes was found to be a hybrid combination

of boxcar and Hamming gates. The effectiveness of hybrid gating over traditional boxcar and semitapered gating is confirmed by an analysis of covariance matrices and error bars. The results show that the developed method effectively suppresses VLF noise in the middle gates, that is, gates with center times spanning 30 $\mu$s to 200 $\mu$s and in the late gates, that is, gates with center times spanning 200 $\mu$s to 1130 $\mu$s. The analysis shows that the average improvement in standard errors obtained for the hybrid gating scheme over boxcar gating is 1.719 and 1.717 on synthetic and field data, respectively.

**1  Introduction**

The Transient Electromagnetic, TEM method is a well-known and well-accepted technology utilized in reservoir exploration, groundwater mapping, saltwater boundary mapping, and a variety of other applications (Auken et al., 2017; Law et al., 2019; Chen et al., 2020). TEM surveys can be done using ground-based instruments that are moved manually between measurements or instruments mounted on mobile platforms such as fixed-wing aircrafts and helicopters, or towed behind all-terrain vehicles

(Mulè et al., 2012; Auken et al., 2019; Liu et al., 2019; Chen et al., 2022). Moving systems are a cost-effective solution with high spatial resolution and allows for large-area mapping. New emerging applications of TEM include permanently installed monitoring instruments that tracks changes in the groundwater levels through daily measurements. However, such applications





greatly increase the need for high Signal-to-Noise Ratio (SNR) data as the groundwater induced variations in the signals are embedded in large background signals (Zamora-Luria et al., 2022).

Multiple noise sources interfere with the TEM signals. The noise sources include powerline noise (50 Hz or 60 Hz) and its harmonics, internal instrumental noise (thermal noise, aging of electronics components, etc.), environmental noise (sferics noise from thunderstorms, traffic, etc.), interference from radio communication systems in the Low Frequency, LF (30-300 kHz) and Very Low Frequency, VLF range (3-30 kHz), and motion noise in the moving systems (Liu et al., 2019; Chen et al., 2022; Macnae et al., 1984; Rasmussen et al., 2018a). Signals from VLF and LF radio communication are one of the most

significant sources of noise for TEM systems. The gating of data is one way to reduce the effect of VLF radios in measured TEM data. Gating has traditionally been carried out with analog boxcar integrators, but new fully-sampled digital receivers offer more flexibility in the gate design. Many different gate shapes with correspondingly different frequency responses are available. However, careful selection of gate shape and placement is necessary to achieve the minimum VLF noise influence in the gated signal. No standard method or principle is available, suggesting a choice of a particular gate shape to suppress the

effect of VLF radios. Selecting the right gate shape and position manually is time-consuming and prone to poor performance due to spontaneously or spatially varying environmental conditions and changing strengths of VLF radios.

    Hence, this work aims to design an optimum gating scheme that is able to suppress the VLF radios in the gated signal and able to adjust itself regularly, e.g., to track daily changes in noise conditions in monitoring instruments. Its significant contributions are:

– Design and implementation of a fully sampled 4 MHz synthetic model for generating noise-less and noisy TEM signals.

    – Design of gate banks with selected gate shapes to be utilized for gating TEM signals from moving or stationary instruments.

    – Implementation of a new covariance matrix, CM tool for studying the presence of VLF noise in the data.

    – Design of a hybrid gate distribution scheme for suppressing the effect of VLF noise in measured TEM signals.

– Optimization of gates by minimizing a multi-objective cost function composed of standard errors of gated data, signal distortion, and covariance.

    – Validation of the hybrid gating model on synthetic TEM signals and field data acquired in Denmark with an analog boxcar gating instrument.

    The remainder of the paper is structured as follows: Section 2 provides a background of the TEM system, related work, and

motivation. Section3 provides a detailed discussion and implementation of the proposed model. Details about the experimental setup of the synthetic and towed-TEM model are covered in Section 4. The results of the proposed model are discussed in Section 5. Finally, conclusions are presented in Section 6.



## 2 Transient Electromagnetic Systems

### 2.1 Background

Briefly, the principles of TEM measurements are as follows (Nabighian and Macnae, 1991) and as shown in Fig. 1. A primary
magnetic field is generated by applying an input current pulse to a transmitter coil. The coil is normally laid out on the ground
or attached to a moving platform. The current is turned on smoothly and turned off abruptly. Following Lenz's law, eddy
currents are induced in the earth below the coil when the primary field is turned off abruptly. The eddy currents produce a
secondary magnetic field, which is then measured by a receiver coil (an induction coil). The eddy currents decay and the shape
of the decay is controlled by the resistivity of the earth, which can therefore be retrieved from the data. The measured response
following a single transmitter pulse is termed a transient. Details of the earth's subsurface resistivity structure are encoded in
the secondary magnetic field, and can be extracted by an inversion procedure (Nabighian and Macnae, 1991). As seen from
Fig. 1, the primary field consists of positive and negative pulses. Sequences of alternating sign pulses are used to mitigate the
effect of powerline harmonics and instrument bias by sign correction and stacking of data (Macnae et al., 1984). The resulting
TEM signals have a high amplitude and bandwidth in the early times, but both amplitude and bandwidth drop rapidly towards
the late times, approximately as $t^{-5/2}$, resulting in lower values of the SNR at late times.

The high-frequency content in the early times is captured by selecting short early gate widths, typically on the order of a
few $\mu$s. In comparison, the signals' late times response and low-frequency content are captured by gates of increasingly longer
duration. Gating is the integration of the signal over specified adjacent time-windows. Essentially, gating provides a decimation
of data, and it corresponds to a filtering of the signal with a filter whose frequency response is obtained by taking the Fourier
transform of the gate shape. As a result, gating is critical for signal interpretation and data reduction.

### 2.2 Related Work

Many studies have been conducted to investigate the properties of VLF radio signals in various fields, including lightning-
ionosphere interactions and unmanned airborne geophysical surveys. To study the effect of lightning on the ionosphere, a
technique based on probing sub-ionospheric VLF has been investigated (Inan et al., 2010). VLF measurements using airborne
systems have been used for geophysical surveys to investigate ground conductivity (Eppelbaum and Mishne, 2011; Pedersen
and Oskooi, 2004). Similarly, the VLF signals have also been used for mapping the changes in the conductivity of the sedi-
mentary covers (Oskooi and Pedersen, 2005). It should be noted that the methods discussed above are concerned with studying
or using the effect of VLF radios in measurements rather than suppressing them.

The role of VLF noise in the measurement and analysis of TEM signals is significant. Boxcar gating with a rectangular
window using analog integrators has been used widely. However, the frequency response of the boxcar has large side lobes
that can let the VLF radios in the measured signal pass through (Harris, 1978). If the TEM instrument allows for acquisition
of many short boxcar gates during single transients, semi-tapered gates with improved frequency responses can be formed as
linear combinations of the short gates, which leads to improved suppression of VLF radios (Larsen et al., 2021).



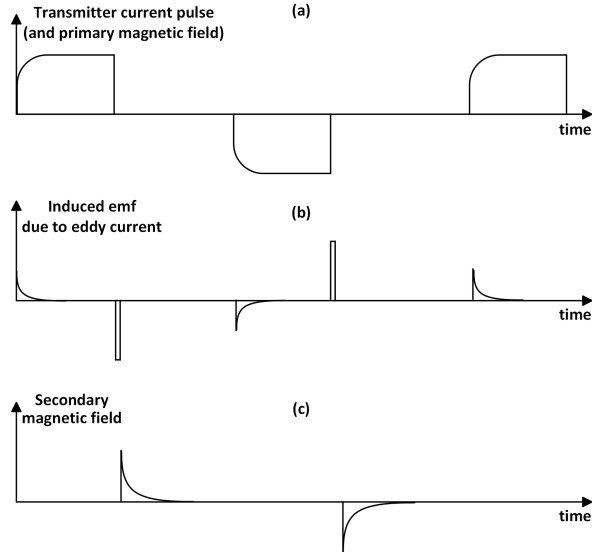

**Figure 1.** Exemplary waveforms of the TEM system; (a) Current pulse to transmitter coil, (b) induced electromotive force from eddy current, and (c) decaying secondary field in receiver coil.

New full-sampled digital receiver systems can employ any desired gate shape, e.g., a B-spline based gating scheme has been explored to compress the measured TEM data in the SkyTEM receiver system (Nyboe and Mai, 2017). The splines are distributed such that at any instant of time, a unity weight is obtained after summing the respective weights of spline coefficients. In a recent paper, the suppression of VLF radios by B-spline gating has been further examined (Peng et al., 2022).

    However, the semitapered and B-spline-based gating schemes have only used an optimization metric about reducing standard
error bars to suppress the effect of VLF radios (Peng et al., 2022; Larsen et al., 2021). Since the standard error bars of the measured data are inversely proportional to the width of the gates, increasing the gate width improves the SNR, but does not guarantee suppression of VLF radios. VLF radio signals remain coherent across individual TEM measurements; therefore, mere stacking does not necessarily suppress the underlying VLF radios. In fact, due to coherence between the repeating TEM signals and VLF radio signals, there is a possibility of lower error bars in the measured data after gating; yet VLF radio
signals are still present in the data (Larsen et al., 2021; Peng et al., 2022). A subtraction method to remove the VLF radios from measured TEM data has been presented in (Macnae, 2015; Rasmussen et al., 2018b). The subtraction method works by identifying the bit streams in the VLF radio signals. The bit streams are then used to generate a copy of the VLF radio signals, which can be subtracted from the received TEM signals. However, this is a computationally expensive method compared to gating. Importantly, only a few papers have been published about suppressing VLF noise in TEM measurements. This suggests
that further investigations could potentially lead to improved SNR.



## 2.3 Motivation

During the late times, from about 100 $\mu$s and onward, the VLF signals' amplitude dominates the TEM signal's amplitude. Due to the VLF contribution and the unintended coherence between the repeating TEM signals and the VLF interference, the resultant TEM signal can falsely appear oscillating and with low standard errors (Macnae, 2015). Therefore, merely minimizing standard error bars does not guarantee suppression of VLF and an optimization metric based only on lowering standard errors might not be a sufficient tool for reducing VLF interference. However, an analysis of the covariance between the gates can identify the presence of VLF interference. Minimization of the covariance can therefore be used as a second optimization criteria. The proposed method further introduces a requirement of minimum distortion of the TEM signals during gating.

Hence, suitable gating methods will increase system performance by increasing SNR, i.e., reduced error bars, will reduce the correlation between different gates i.e., suppressing the effect of VLF and minimize the distortion of the TEM signal by the gating. Normally the same shape is used for all gates, but the paper investigates the potential gains obtained by allowing for an individually selected shape for each gate. To reduce the size of the search space, the shapes are selected from a bank of eight predefined shapes.

## 3 Proposed system model

The proposed synthetic model and the optimal and hybrid gating schemes are described in detail in this section. The first part presents the fully sampled synthetic model for generating noiseless and noisy TEM signals. The second part contains a gate bank with its mathematical formulations. Finally, using Multi-Objective Particle Swarm Optimization, MO-PSO, the paper presents design of optimum gate selection and hybrid gate distribution in the third part.

### 3.1 A Fully Sampled Synthetic Transient Electromagnetic Model

The developed fully sampled synthetic model for TEM signals is composed of three types (a) pure TEM signals, (b) TEM signals with Random Noise, RN, and (c) TEM signals with RN, powerline harmonics, and VLF noise. The proposed model is designed to mimic the characteristics of the towed-TEM instrument (Auken et al., 2019).

#### 3.1.1 Transient Electromagnetic Signal

The timescale for measuring TEM signals ranges from microseconds to a few tens of milliseconds, covering a wide dynamic range. The TEM signal decays approximately as $t^{-5/2}$ (Nabighian and Macnae, 1991). Gating the TEM signal over time produces an output that decays at a rate of $t^{-3/2}$, reducing the need for dynamic range in analog to digital conversion.

#### 3.1.2 Powerline Noise

Electrical powerlines generate noise from an electric and magnetic field at the fundamental frequency of transmitted power (50 Hz for Denmark) and its harmonics. The frequency, amplitude, and phase of the powerline harmonics vary with time due



to load and demand in the power grid, but are approximately constant for short data records, usually a few seconds or less. The powerline or harmonic noise with a fundamental frequency of $f_p$ is represented by

$$p_l(t) = \sum_{i=1}^{i_{max}} A_{P_i} sin(2\pi i f_p t + \phi_i) \tag{1}$$

where $A_{P_i}$ and $\phi_i$ is the amplitude and phase of the fundamental frequency and their harmonics, $f_p$ is the fundamental frequency, and $i_{max}$ is the maximum number of harmonics.

### 3.1.3   White noise

TEM signals are affected by broadband noise, which stems from thermal noise in the receiver coil and amplifier electronics, along with atmospheric noise. The paper models this broadband noise as a stationary white noise Gaussian process (Kuo, 2018).

### 3.1.4   Very Low Frequency Radios

Signals from VLF radios are generated using antennas spread across the world, causing interference with TEM systems. Here these VLF radio signals are generated as random bit streams and encoded using Minimum-Shift-Keying, MSK. The frequency content of these VLF radios overlaps with the frequencies of TEM signals resulting in reduced data quality (Zhang, 2015). Table 1 shows some significant VLF radios (Cohen, 2006; Kavanagh et al., 2011). Details of MSK and Gaussian MSK (GMSK) encoding can be found in (Rasmussen et al., 2018b).

We model the TEM signal as an alternating pulse train along with powerline harmonics, RN, and VLF radio noise to create the final noisy TEM system model. The mathematical formulation for the proposed synthetic model is given as

$$s(t) = \sum_{k=0}^{K-1} (-1)^k A_T((t-kT_0)^{-5/2})U(t-kT_0)...$$
$$... + A_{RN}x(t) + A_{VLF}z(t) + p_l(t) \tag{2}$$

where $A_T$ is the amplitude of the decaying TEM signal, $A_{RN}$ and $A_{VLF}$ are amplitudes of the RN and VLF radios. The constant $A_T$ is directly proportional to the current in the transmitter loop, electrical conductivity, transmitter/receiver coil area,
and frequency-independent magnetic permeability (Christiansen et al., 2009). The $U(t)$ is the unit step function with a shift of $kT_0$, where $k$ is the number of transients and $T_0$ is the pulse repetition time, $x(t)$ is the RN, and $z(t)$ is the VLF radio signals generated using MSK or GMSK.

In practice, the powerline frequency is constant over a small duration. The correct choice of $T_0$ ensures that the maxima and minima of the powerline noise are coherent with the TEM measurement. Therefore, synchronous detection, i.e., sign correction
of the alternating sign pulses and subsequent stacking of the TEM signals, results in the cancellation of powerline noise and its harmonics. In countries with different powerline frequencies, normally 60 Hz, the repetition rate ($T_0$) is changed to match the powerline frequency such that it results in cancellation after sign correction. The resultant TEM signal without noise and with





**Table 1.** Examples of VLF radio stations (Cohen, 2006; Rasmussen et al., 2018b), see also https://en.wikipedia.org/wiki/List_of_ VLF-transmitters. Most stations use the MSK format and operates at 200 bits per second.

| F (kHz) | Call | Power (MW) | Location |
|---------|------|------------|----------|
| 19.8 | NWC | 1.0 | Exmouth, Australia |
| 21.75 | HWU | 0.4 | Rosnay, France |
| 23.4 | DHO | 0.8 | Rhauderfehn, Germany |
| 24.0 | NAA | 1.8 | Cutler, Maine, USA |
| 24.8 | NLK | 1.2 | Seattle, Washington, USA |

noise is shown in Fig. 2. As negative values plotted on a log scale are ignored, these values are represented by a square marker by taking the absolute values. The magnified version of the TEM signal is shown in the upper box of Fig. 2 (b) and (c).

## 3.2 Gate Bank

The current work selects standard gates based on their frequency, in particular side lobe properties. However, many other gate shapes can also be employed, but their properties are generally quite similar.

### 3.2.1 Boxcar Gate

The rectangular or boxcar gate is an array of one defined for a desired interval in time. The mathematical formulation of a boxcar gate is defined as (Harris, 1978)

$$g(t) = \begin{cases} 1, & 0 \leq t \leq T \\ \\ 0, & \text{otherwise.} \end{cases} \tag{3}$$

### 3.2.2 Hanning Gate

The Hanning gate has a shape like one half cycle of the cosine wave with a DC shift of 1, so it always remains positive. It is expressed as (Harris, 1978)

$$g(t) = \frac{1}{2}\left[1 - \cos\left(2\pi\frac{t}{T}\right)\right] \quad \text{for } 0 \leq t \leq T. \tag{4}$$





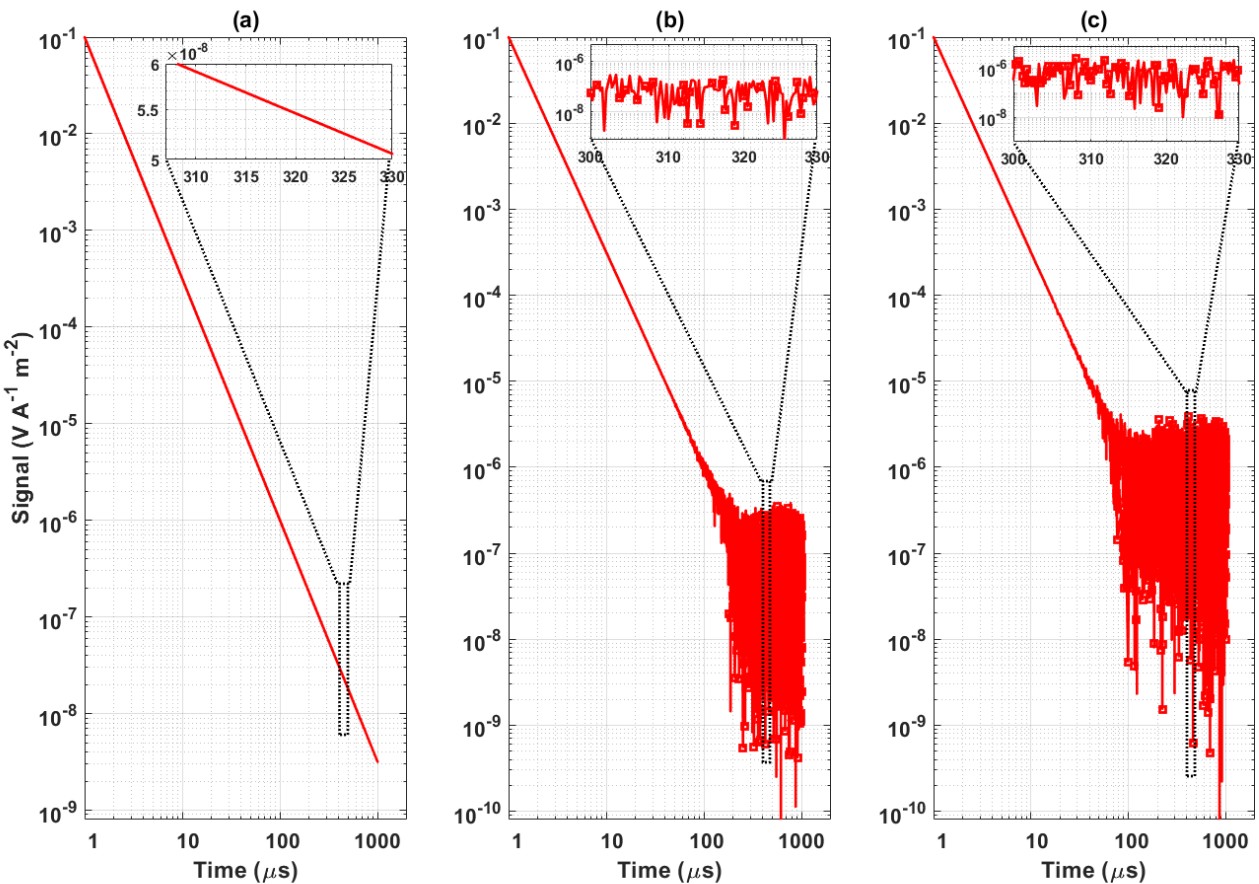

**Figure 2.** Example of a decaying TEM signal. (a) Noiseless TEM, (b) TEM signal and random noise, and (c) TEM signal with random noise and VLF noise. The absolute values of negative samples of the noisy TEM signal are plotted and indicated with a square marker.

### 3.2.3 Hamming Gate

The Hamming gate or window is a tapered gate formed by using a raised cosine like the Hann window, but with a non-zero start and end points. It is also called the tapering or apodization function formulated as (Harris, 1978)

$$g(t) = \left[0.54 - 0.46\cos\left(2\pi\frac{t}{T}\right)\right] \quad \text{for } 0 \leq t \leq T. \tag{5}$$

### 3.2.4 Kaiser Gate

The Kaiser gate, also known as the Kaiser-Bessel window, approximates the prolate spheroidal window in which the ratio of the main lobe to the side lobe power spectral density is maximized. The attenuation in the side lobes is controlled by a tuning





parameter $\beta$. Higher values of $\beta$ increase the width of the main lobe and decrease the amplitude of the side lobes, thereby increasing the attenuation. The Kaiser window is defined as (Harris, 1978)

$$g(t) = \frac{I_0\left(\beta\sqrt{1 - \left(\frac{(t-T)/2}{T/2}\right)^2}\right)}{I_0(\beta)} \quad \text{for } 0 \leq t \leq T \tag{6}$$


where $I_0$ is the zeroth-order modified Bessel function of the first kind.

### 3.2.5 Gaussian Gate

The Gaussian gate is called a bell curve gate and is the only function that Fourier transforms itself with a smooth, non-zero function in the closed form. The mathematically Gaussian gate is denoted as (Harris, 1978)

$$g(t) = \exp\left[-\frac{(t-T/2)^2}{2\sigma^2}\right] \quad \text{for } 0 \leq t \leq T. \tag{7}$$


### 3.2.6 Tukey's or Fully tapered Gate

Tukey's gate is a tapered cosine gate whose first and last edge follows a cosine shape while the central follows the rectangular window. It is also called a fully tapered gate (Larsen et al., 2021). The mathematical expression defining the Tukey's gate is defined as (Harris, 1978)

$$g(t) = \begin{cases} \frac{1}{2}\left[1 + \cos\left(\frac{2\pi}{t_1}\left(t - \frac{t_1}{2}\right)\right)\right], & 0 \leq t < \frac{t_1}{2} \\ 1, & \frac{t_1}{2} \leq t < T - \frac{t_1}{2} \\ \frac{1}{2}\left[1 + \cos\left(\frac{2\pi}{t_1}\left(t - T + \frac{t_1}{2}\right)\right)\right], & T - \frac{t_1}{2} \leq t \leq T \end{cases} \tag{8}$$


where $t_1 = mT$ and $m$ represents the fraction of the window inside the cosine tapered region. The current work used a value of $m = 0.5$, producing a Tukey window where 1/2 of the entire window length consists of segments of a phase-shifted cosine with period $2m = 1$. A value of $m = 0$ represents the rectangular or boxcar gate, and $m >= 1$ is equivalent to the Hanning gate.

### 3.2.7 Semitapered Gate

Here, a semitapered gate is defined as a piece-wise flat approximation of the cosine part of the fully tapered or Tukey's window gate (Larsen et al., 2021).


### 3.2.8 B-Spline Gate

The B-spline gate is formulated using the Cox-de-Boor recursive formula such that it is a composite curve of degree $p$ polynomials, provides local support, and forms a partition of unity. The second-order B-spline function formulated is represented by (Nyboe and Mai, 2017)

$$
g(t) = \begin{cases}
\frac{(t-t_0)^2}{(t_2-t_0)(t_1-t_0)}, & t_0 \leq t < t_1 \\[2ex]
\frac{(t-t_0)(t_2-t)}{(t_2-t_0)(t_2-t_1)} + \frac{(t-t_1)(t_3-t)}{(t_3-t_1)(t_2-t_1)}, & t_1 \leq t < t_2 \\[2ex]
\frac{(t_3-t)^2}{(t_3-t_1)(t_3-t_2)}, & t_2 \leq t < t_3 \\[2ex]
0, & \text{otherwise}
\end{cases}
\tag{9}
$$

where $t_0, t_1, t_2$, and $t_3$ are the time knots selected such that they are equally spaced in time. The resultant spline is a combination of three parabolas such that at any instant of time, the sum of weights for all parabolas active is unity. The gate shapes presented in the above subsection are in continuous time. In the current work, the piece-wise flat gates have been constructed with a weighted combination of short analog boxcars called sub-gates.

### 3.3 Optimized and Hybrid Gating Scheme

The suppression of VLF noise is accomplished by solving a multi-objective optimization problem such that a cost function composed of gate standard error bars, covariance between gates, and distortion of the gated TEM signal is minimized by varying the gate shape and gate width. Different gate shapes are inputted to the optimization problem to get the optimal solution. The multi-objective cost function used for obtaining the optimal gating scheme is defined as

$$
(g_{opt}, P_{opt}) = min(mean(E), mean(C), mean(D))
\tag{10}
$$

where $g_{opt}$ is the optimum gate shape, $P_{opt}$ is the proportion of overlapping with the adjacent gate, $E$ is standard error bars, $C$ is the normalized covariance terms between gates and $D$ is the distortion of the noisy gated TEM signal. The gate locations are fixed because gate widths are designed so that they are exponentially widening. The current work considers a set of measurement composed of $N_t$ transients, each containing $N_g$ gated values. The data from the $i$'th transient is stored in a $N_g \times 1$ vector

$$
\boldsymbol{s}_i = \begin{bmatrix} s_{g_1} & s_{g_2} & \cdots & s_{g_{N_g}} \end{bmatrix}^T.
\tag{11}
$$

The stacked (averaged) signal is computed as

$$
\bar{s} = \frac{1}{N_t} \sum_{i=1}^{N_t} \boldsymbol{s}_i.
\tag{12}
$$





Using this, the mean-removed data matrix is formed with size $N_g \times N_t$ as

$$S = \begin{bmatrix} \boldsymbol{s}_1 - \bar{s} & \boldsymbol{s}_2 - \bar{s} & \dots & \boldsymbol{s}_{N_t} - \bar{s} \end{bmatrix}^T.$$ (13)

From which $N_g \times N_g$ covariance matrix is formed which is denoted as

$$CM = \frac{1}{N_t - 1} SS^T.$$ (14)

Due to the high dynamic range of TEM signals, which spans multiple orders of magnitude, careful normalization is required
to ensure that all gates are evenly contributing in the optimization.

The diagonal of the covariance matrix contains the variance of each gate. From this we form the $N_g \times 1$ vector $E$ of signal-normalized standard errors with the elements $E_i = CM_{ii}/\bar{s}_i$. The off-diagonal terms are normalized by dividing the $ij$'th element by $\sqrt{CM_{ii}CM_{jj}}$, which therefore results in Pearson's linear correlation coefficient. The mean of $C$ used in the optimization is computed from the absolute value of the normalized off-diagonal terms.

Finally, a signal-normalized measure of the distortion of the TEM signal is computed by forming the $N_g \times 1$ vector $D$ as

$$D = \left| (I \boldsymbol{s}_{ideal})^{-1} (\boldsymbol{s}_{ideal} - \boldsymbol{s}_{noise}) \right|$$ (15)

where $\boldsymbol{s}_{ideal}$ is the ideal gated TEM signal without any noise, modelled as a $t^{-5/2}$ decay, $\boldsymbol{s}_{noise}$ is the gated noisy TEM signal and $I$ is the identity matrix.

The MO-PSO is an evolutionary meta-heuristic optimization algorithm that has been used in various engineering and other
applications (Coello Coello and Lechuga, 2002; Reyes-Sierra and Coello, 2006; Yang et al., 2022). The generic MO-PSO is to (Yang et al., 2022)

$$minimize([f_1(w), f_2(w), ... f_k(w)]).$$ (16)

Subject to the constraints of (Yang et al., 2022)

$$h_i^1(w) \leq 0 \quad i = 1, 2, ..., p$$

$$h_i^2(w) = 0 \quad i = 1, 2, ..., q$$ (17)

where $w$ is a set of decision arrays, which is $(g, P)$ in this work, $f_k$ are the objective functions, here $(mean(E), mean(C), mean(D))$, $k$ is the number of objective functions, and $h_i^1$ and $h_i^2$ are the constraint functions of the problem. The particle positions are iteratively updated with (Yang et al., 2022)

$$w_{i,j}(m+1) = w_{i,j}(m) + v_{i,j}(m+1)$$

$$v_{i,j}(m+1) = \psi \times v_{i,j}(m) + d_1 \times (p_{b_{i,j}}(m)...$$ (18)
$$... - w_{i,j}(m)) + d_2 \times (g_{b_{i,j}}(m) - w_{i,j}(m))$$





where $P_{b_i}(m) = (p_{b_{i,1}}(t), p_{b_{i,2}}(t), ... p_{b_{i,D}}(t))$ is the personal best of each particle in search space obtained so far, $G_{b_i}(m) =$
$(g_{b_{i,1}}(t), g_{b_{i,2}}(t), ... g_{b_{i,D}}(t))$, is the global best obtained so far, $\psi$ is the parameter that controls the search space of the particle's
exploration called as inertia weight, $d_i$ is a scalar product of $a_i$ and $c_i$ with $c_1$ and $c_2$ are non-negative constants, while the
values of $a_1$ and $a_2$ are within [0,1], and $m$ is the current iteration. The variable $w_{i,j}(m+1)$ corresponds to the update in
position (from search space), i.e., update in the position and type of gate, whereas $v_{i,j}(m+1)$ is the rate of velocity of the $i$th
particle. The proposed work uses an MO-PSO algorithm to select the desired gate and its placement accurately. The variable
$w_{i,j}(m)$ is the possible gate shape and its position, the next position of the particle for $m + 1$ instance is obtained by adding
its velocity $v_{i,j}(m)$ to $w_{i,j}(m)$. After each iteration, the personal best of each particle is obtained from a pool of search agents
with the local best values for gate shape and position. Once the maximum number of specified iterations is completed, the
global best solution is obtained, with the minimum values of objective functions. The randomness of the starting points in
the MO-PSO search for the optimal solution, implies that the algorithm can get trapped in a false minima. To overcome this
limitation, the proposed work introduces confined search spaces where the first and second half of the gates are each confined
to only one shape. The current work denotes this confinement with two shapes as hybrid gating.

## 4  Experimental Setup

### 4.1  Synthetic Model

The fully sampled synthetic TEM model is implemented in MATLAB (R2021a) on a 64-bit Windows operating system with an
Intel(R) Core(TM) i7-8550U, 1.80 GHz CPU system with 16 GB RAM. The current work generates the TEM, VLF, powerline
interference, and RN at a 4 MHz sampling frequency. A total of 1000 TEM signals with a 1 ms duration are generated, decaying
at a rate of $t^{-5/2}$ ($\frac{V}{Am^2}$).

In the VLF model, a bandwidth of 100 Hz ($\Delta f$) for all radio signals is selected (Macnae, 2015). The model is designed such
that for any user-defined number of VLFs, the system chooses random VLF radios to remove the bias between the choice and
design of the VLF radio shown in Table 1. The signal record lengths are chosen so that stacking cancels powerline noise and
its harmonics. The values of $A_{RN}$ and $A_{VLF}$ in Eq. (2) are selected such that distortions in resultant TEM signals start after
100 $\mu$s (Larsen et al., 2021). Three scenarios are used on the synthetic TEM data (*TEM+RN+VLF radios*), i.e., with one VLF
radio, with four VLF radios, and with eight VLF radios for the analysis. The 4 MHz data stream is assembled into 84 boxcar
gates to mimic the experimental conditions (Larsen et al., 2021). The boxcar gates are spaced approximately uniformly on a
logarithmic time scale, except for the first ten gates, which are distributed linearly in time, enforced by the 0.25 $\mu$s sampling
period. Subsequently, the 84 gates are re-gated, i.e., linearly combined, into 30 gates with the re-gating being determined from
the gate bank and optimization function. These 30 gates, denoted as production gates, are the final output, which can be used
for inversion. The first gate of the final gate distribution is centered at 0.6 $\mu$s with 1 $\mu$s width, and the last gate is centered
at 974.17 $\mu$s with 51 $\mu$s width. The first four gates of the final gates are distributed linearly and uniformly in time, while the
rest of the gates are spaced uniformly on a logarithmic time scale. The first decade comprises eight gates, while the remaining





decades contain 11 gates per decade. The weights of the final gates are normalized with respect to the raw gate width of analog boxcar gates to maintain the TEM signal value.

## 4.2 Field Data

The field measurements have been recorded using a ground-based monitoring system based on analog boxcar gating (Zamora-
Luria et al., 2022). The system consists of a transmitter coil (2 m × 4 m) and a receiver coil (0.5 m × 0.5 m), that are placed 7 m apart from the transmitter coil. Measurements of shallow and deep geological data have been accomplished using dual moments (low and high current transmitter pulses) corresponding to early and late-time TEM recordings. Here, pulse repetition rates of 2110 Hz and 630 Hz are used for low and high-moment data recordings. Each low and high moment stack comprises 422 and 252 transients, respectively. A detailed system configuration can be found in (Auken et al., 2019). Here, the focus is
on the high moment data with 84 gates spanning from 0.3 $\mu$s to 1130 $\mu$s, where the effect of VLF noise at late gate times can be significant.

## 5 Results

The section presents the results obtained for the proposed model on synthetic and field data. In particular, the presence of VLF noise in the synthetic data is examined by the patterns produced in the CM and the improvement by the optimized gating
scheme is examined. Fully sampled data generated at the 4 MHz sampling frequency are assembled into 84 boxcar gates and used to form the CM in various noise scenarios, Fig. 3. When TEM signals are not affected by any VLF noise, gates are highly non-correlated, and the CM is nearly diagonal, as shown in Fig. 3 (a). When VLF noise is present in the data, the VLF noise decides the CM pattern. With multiple VLF radios operating at different amplitudes and frequencies, the resultant CM has dense stripes representing the higher correlation between the gates, Fig. 3 (b), (c), and (f). For fewer VLF radios, the patterns
are less strong, Fig. 3 (d) and (e). Generally, with varying amplitude and combinations of VLF radios, the stripes pattern on the CM also varies. Therefore, this example demonstrates that correlation analysis using a CM can be used as a new tool to detect the presence of VLF interference in TEM data.

### 5.1 Synthetic Data Analysis

Fully sampled data generated at the 4 MHz sampling frequency are assembled into 84 boxcar gates. After obtaining the 84
boxcar gates, they are re-gated to 30 gates, where the MO-PSO is used to search for the best gate shape or combinations of gate shapes such that it minimizes the cost function. The optimization was tested for three different scenarios, i.e., TEM signals with random noise and either one, four, or eight VLF radios. One thousand transients have been used for each scenario to find the optimal gate combination. Most frequently, a combination of boxcar and Hamming gates was obtained as the best solution, followed by semitapered boxcar and Hamming gates. Other gates have also been found as the best solution for the optimization
problem; however, they occurred few times. From an entire solution space obtained at last, a hybrid combination of only boxcar and Hamming gates has been found to be the best solution for the majority of the transients.



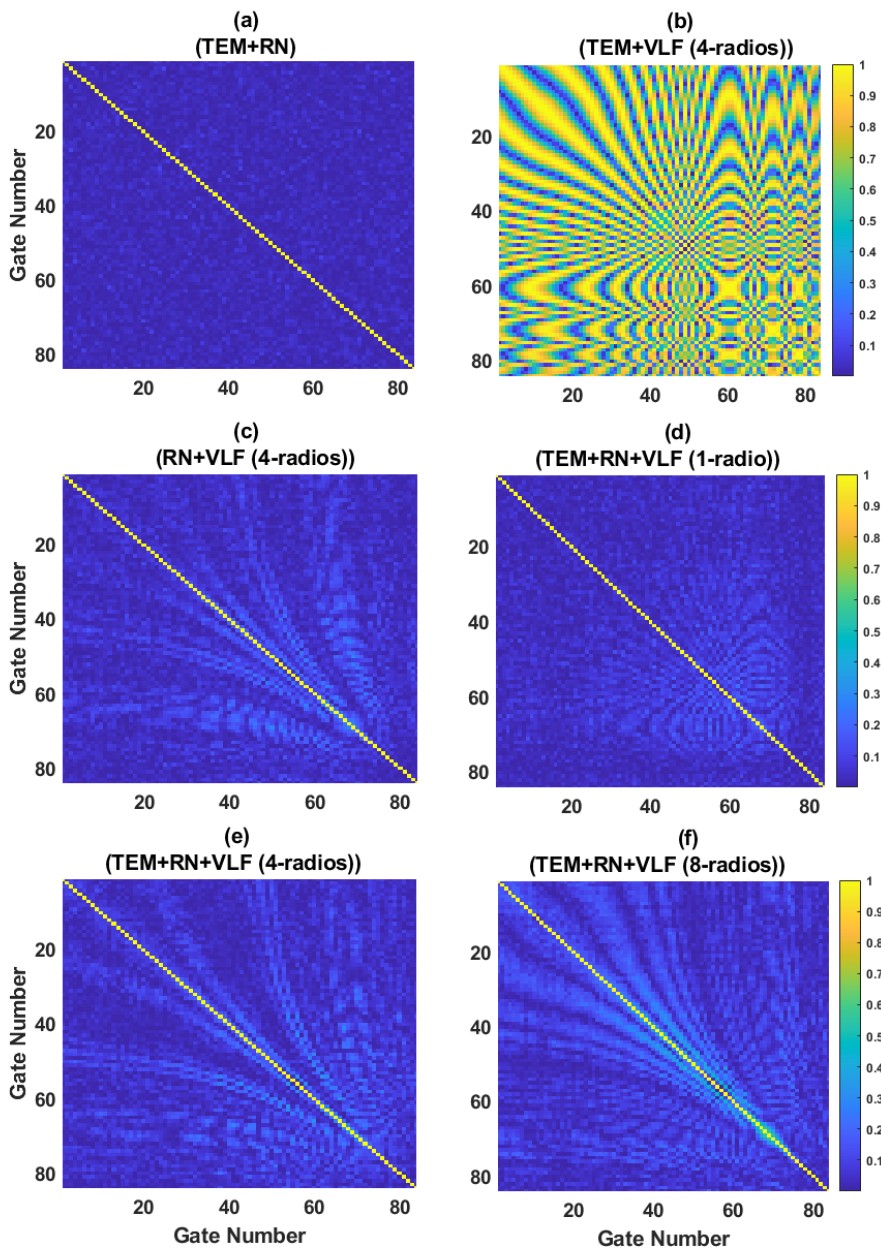

**Figure 3.** The covariance matrices obtained for 84 boxcar gates. (a) TEM and random noise, (b) TEM + VLF noise, (c) random noise + VLF noise, (d) TEM with random and VLF noise (1 VLF radio), (e) TEM with random and VLF noise (4 VLF radios), and (f) TEM with random and VLF noise (8 VLF radios).





The result of the final gate combination is such that for early times, gate number 1-10, from 0.3 $\mu$s to 30 $\mu$s and mid-early times, gate number 11-15, from $30\mu$s to 80 $\mu$s, boxcar gates provide the best suppression in the VLF with low correlation and negligible standard error bars. During mid-late times, i.e., gate number 16-20, from 80 $\mu$s to $200\mu$s and late times, i.e., gate

21-30, from $200\mu$s to 1 ms, the Hamming window gives better performance.

Fig. 4 shows a comparison of the CM obtained using boxcar gates, semitapered gates, and the hybrid gates scheme. The corresponding transients and error bars are shown in Fig. 5. It is evident from Fig. 4 that the boxcar gate suppresses VLF radios during early and mid-times, whereas the effect of the VLF is more significant at late times. When semitapered gating is used, it suppresses VLF radios at late times, but the early and mid-times are significantly affected by the presence of VLF noise.

Analysis of standard error bars on the re-gated data shown in Fig. 5(a) reveals that the boxcar gates provide higher error bars for all three scenarios, i.e., TEM and RN with 1, 4, and 8 VLF radios, respectively. This makes boxcar gates a poor choice during the late gates. The analysis of semitapered gates shows that the error bars produced by them are lower than those of boxcar gates and provide a smoother signal at late times. It has been observed that hybrid gating provides higher suppression of VLF radios than semitapered gates during early times.

In contrast, during the mid and late times, the hybrid scheme suppresses the VLF radios better than boxcar gates. Similarly, the analysis of standard error bars reveals that the hybrid gating scheme provides small error bars in the early, mid, and late times. The covariance and standard error bar analysis confirm that hybrid gating improves suppression of VLF radios. To further investigate the model's performance, this work also evaluates two improvement factors defined as (Larsen et al., 2021)

$$\gamma_{hybrid} = \frac{SE_{boxcar}}{SE_{hybrid}}$$

$$\gamma_{semitapered} = \frac{SE_{boxcar}}{SE_{semitapered}}$$

(19)

which measures the improvement in standard errors using either hybrid or semitapered gating relative to boxcar gating.

Table 2 shows the summary of the improvement factor obtained for gates 15-24 for 1000 TEM signals using semitapered and hybrid gates. It is evident from the table that the semitapered gates have a somewhat higher improvement factor compared to the hybrid gates, but the standard deviation is also higher. However, for gates number 21 and 22, the improvement factor provided by hybrid gates is higher than that of semitapered gates with smaller standard deviation. The effectiveness of the optimized

model is further evaluated by obtaining the mean values of covariance and distortion for noisy TEM signals with different combinations of VLF radios. Tables 3 and 4 show the mean values of covariance and distortion obtained for different gates. It is evident from the Tables that boxcar and semitapered gates trade-off between covariance and distortion. Decreased covariance values with boxcar gates result in higher distortion, whereas lower distortion for semitapered gates results in increased covariance. However, hybrid gates provide an optimal solution with minimal correlation and distortion.



**Figure 4.** Covariance matrices obtained after re-gating for TEM + RN + VLF noise (i) 1 VLF radio, (ii) 4 VLF radios, and (iii) 8 VLF radios with (a) Boxcar gates, (b) Semitapered gates, and (c) Hybrid gates.





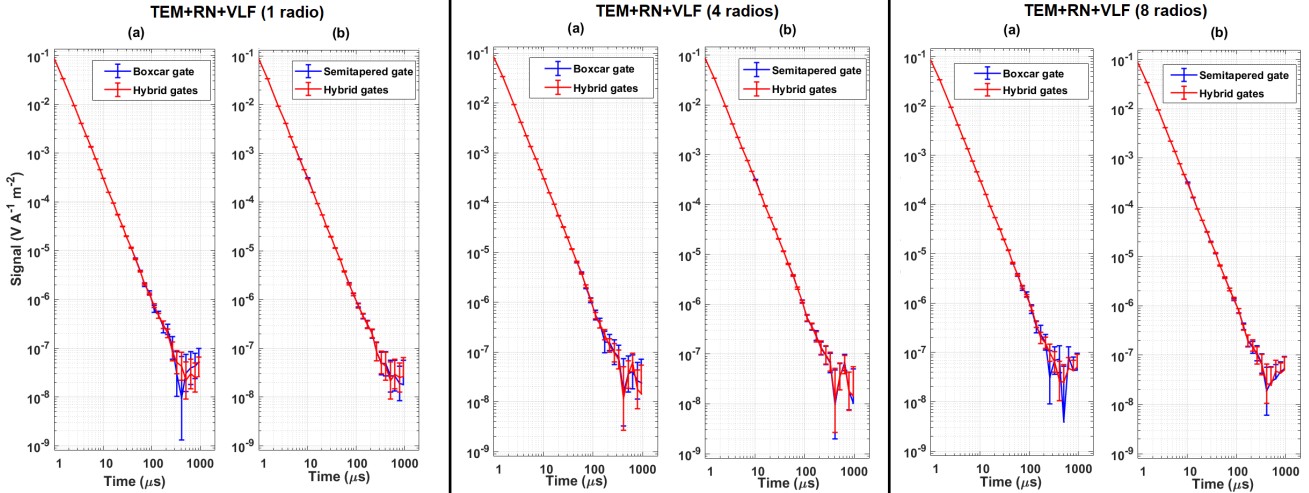

**Figure 5.** Error bars obtained for the re-gated signal composed of TEM with random and VLF noise in the case of 1, 4, or 8 VLF radios (a) Boxcar gate vs Hybrid gates and (b) Semitapered gate vs Hybrid gates.

**Table 2.** Improvement factors and associated standard deviation for gates 15 to 24 on synthetic data.

| Gate No. | Gate Center ($\mu s$) | Improvement Factor ($\gamma_{hybrid}$) | Improvement Factor ($\gamma_{semitapered}$) |
|---|---|---|---|
| 15 | 37.733 | $1.302 \pm 0.005$ | $1.353 \pm 0.007$ |
| 16 | 46.807 | $1.312 \pm 0.010$ | $1.379 \pm 0.013$ |
| 17 | 58.190 | $1.670 \pm 0.025$ | $1.908 \pm 0.037$ |
| 18 | 72.322 | $1.587 \pm 0.039$ | $1.772 \pm 0.054$ |
| 19 | 89.940 | $2.214 \pm 0.109$ | $2.544 \pm 0.159$ |
| 20 | 111.781 | $2.147 \pm 0.187$ | $2.452 \pm 0.268$ |
| 21 | 138.822 | $1.840 \pm 0.166$ | $1.822 \pm 0.209$ |
| 22 | 172.509 | $1.845 \pm 0.326$ | $1.765 \pm 0.394$ |
| 23 | 214.368 | $1.624 \pm 0.307$ | $1.690 \pm 0.360$ |
| 24 | 266.238 | $1.653 \pm 0.607$ | $1.748 \pm 0.885$ |





**Table 3.** Mean value of covariance obtained for non-diagonal elements with different VLF radio combinations.

| Signals/Gates | Boxcar | Semitapered | Hybrid |
|---|---|---|---|
| TEM (1 VLF radio) | 0.0415 | 0.0727 | 0.0496 |
| TEM (4 VLF radios) | 0.0544 | 0.0835 | 0.0549 |
| TEM (8 VLF radios) | 0.0917 | 0.1184 | 0.0852 |

**Table 4.** Mean distortion obtained for gate number 22 with different VLF radio combinations.

| Signals/Gates | Boxcar | Semitapered | Hybrid |
|---|---|---|---|
| TEM (1 VLF radio) | $7.27 \times 10^{-08}$ | $4.64 \times 10^{-08}$ | $4.72 \times 10^{-08}$ |
| TEM (4 VLF radios) | $8.13 \times 10^{-08}$ | $4.63 \times 10^{-08}$ | $4.69 \times 10^{-08}$ |
| TEM (8 VLF radios) | $9.49 \times 10^{-08}$ | $4.74 \times 10^{-08}$ | $4.83 \times 10^{-08}$ |

## 5.2 Field Data Analysis

The proposed hybrid gating model is also tested on the field data acquired from Kompedal Plantage, Denmark. A total of 313 signals have been used to analyze the effect of VLF noise on the gated data. The readings are measured using a buried monitoring TEM system. Each signal is a stacked combination of 252 transients of high-moment data containing 84 gates. All 252 transients are re-gated for each signal using the boxcar, semitapered, and the optimized hybrid gate model. After gating all the transients of each signal, they are stacked to form a $313 \times 30$ signal matrix.

Fig. 6 (a) shows the CM obtained for the 84 gates signal, and the other panels show the re-gated (b) boxcar, (c) semitapered, and (d) hybrid gating results. Fig. 7 illustrates the standard error bars obtained on the final gated signal using boxcar gates, semitapered gates, and the hybrid gating scheme. From Fig. 6 (a), it is seen that some stripes confirm the presence of VLF radios, whereas there are also some strong patterns in very early and late times, due to which a box-type structure is visible in the CM. The source of this structure is unidentified and may be due to some instrumental disturbances. It is evident from Fig. 6 (b) and Fig. 7 (a) that the boxcar gates result in a very strong pattern in CM in mid and late times and, therefore, provide higher error bars at late times. In contrast, the hybrid gating scheme suppresses off-diagonal terms in early and mid-times compared to the boxcar and semitapered gates and provides a very smooth decay of the TEM signal compared to a boxcar. Further, Fig. 7 shows that the hybrid gates provide lower values of standard error bars than boxcar gates.

Table 5 shows the summary of the improvement factor obtained on field data for gates 15-24 for 313 signals using semitapered and hybrid gates with respect to boxcar gates. Similar to the simulations, the higher improvement factor provided by the semitapered gates comes with a higher standard deviation. Also, a higher improvement factor in the time-domain analysis does not guarantee suppression of VLF in the frequency domain, as seen in the case of the semitapered gate. The mean values of covariance are obtained as 0.1538, 0.1894, and 0.1711 for boxcar, semitapered, and hybrid gates, respectively. Due to prac-





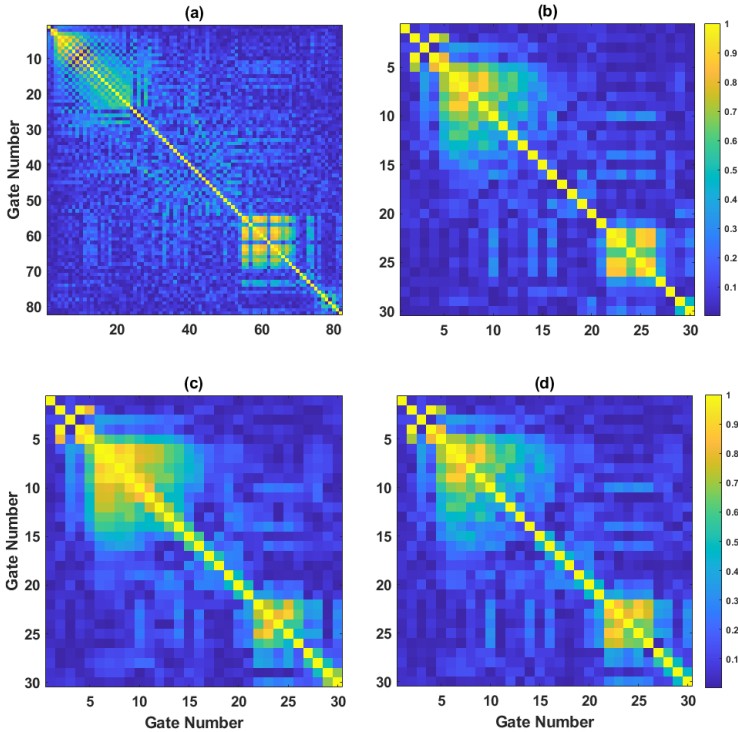

**Figure 6.** Covariance matrices of field data before and after re-gating. (a) Raw gates, (b) Boxcar re-gating, (c) Semitapered re-gates, and (d) Hybrid re-gating.

tical constraints, i.e., the impossibility of obtaining the ideal decay from the instruments on the field data, it is not possible to measure distortion on field data. The analysis reveals that the existing semitapered gates that reduce standard error bars with better improvement factors have a high correlation between the gates, which is reflected in the CM. On the other hand, the hybrid gate combination has not only provided an improvement over boxcar gates in the time-domain analysis by minimizing the standard error bars, but also significantly suppressed the VLF in the mid and late times. Thus, the hybrid scheme is a better

choice for gating the TEM data because it not only provides smaller error bars and reduced correlation between the gates than boxcar gates at late times, but also reduces the correlation of gates in early and mid-times compared to the semitapered gates.

**5.3 Frequency Response Analysis**

Figure 8 (a) illustrates the time-domain gate shapes while (b) shows the corresponding frequency-domain response for boxcar, semitapered, and piece-wise flat Hamming gates. Notably, each tapered and squared part of the semitapered gate comprises

four sub-gates, while for the Hamming gate it comprises of five sub-gates. The gates are designed to appear symmetric on a logarithmic time axis, and will therefore appear asymmetric on a linear time axis. The frequency response is normalized to a unit gain at 0 Hz. It is clear from Fig. 8 that the frequency response provided by boxcar gates has significant side lobes, which





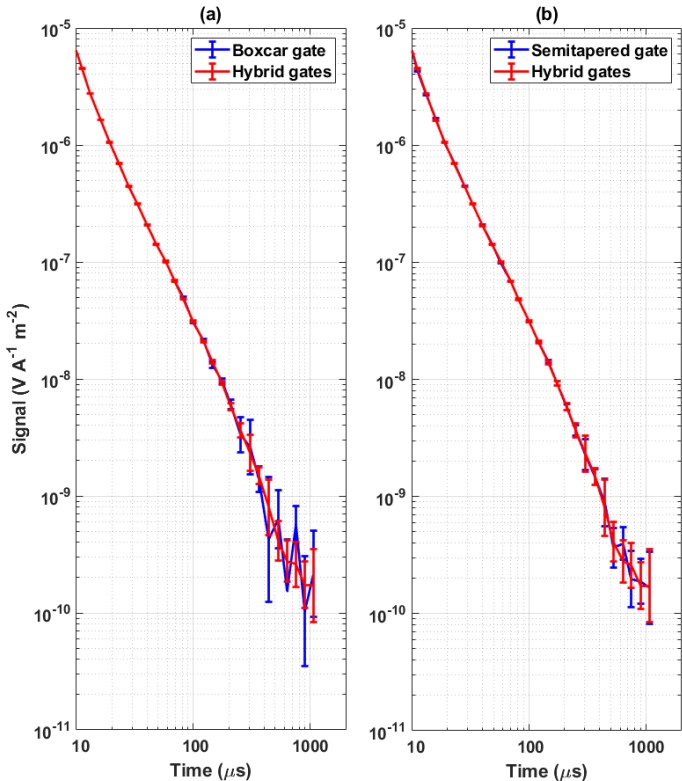

**Figure 7.** Error bars obtained for the re-gated signal using field data (a) Boxcar gate vs Hybrid gates and (b) Semitapered gate vs Hybrid gates.

can pass the VLF radios. The frequency response offered by the semitapered gate provides better suppression of the VLF radios due to smaller amplitudes of the side lobes. The arrows represent the location of common VLF radios in the frequency spectrum. It is seen from the frequency response of semitapered, Hamming and boxcar gates that the later side lobes can pass the VLF radios. The frequency response of the semitapered and Hamming gates offers a very low amplitude of early and mid-side lobes resulting in stronger suppression of VLF radios.

The total decay time of one TEM signal can roughly be divided into three regions viz early times i.e., between gates 1 to 10 spanning from 0.3 $\mu$s to 30 $\mu$s, mid times between gates 11 to 20 spanning 30 $\mu$s to 200 $\mu$s, and late times i.e., between gates 21 to 30 spanning 200 $\mu$s to 1130 $\mu$s. At early times, VLF frequencies appear inside the main lobe due to smaller gate widths (a few $\mu$s). However, the amplitude of the TEM signal dominates over VLF radio noise having a very high SNR; therefore, the effect of the VLF is not dominant, resulting in smooth and undistorted decays of TEM signals. At mid-times, the edge of the main lobe and the first side lobes align with the frequency of VLF radios. Due to this, VLF radio signals can be present in gated data. However, if the amplitude of TEM signals is higher than that of the VLF radios, the TEM signal decays smoothly. On the other hand, if the amplitudes of VLF radios are dominant over signal amplitude, there might be distortion in the resultant gated





**Table 5.** Improvement factors and associated standard deviation for gates 15 to 24 on field data.

| Gate No. | Gate Center ($\mu s$) | Improvement Factor ($\gamma_{hybrid}$) | Improvement Factor ($\gamma_{semitapered}$) |
|---|---|---|---|
| 15 | 69.023 | $1.641 \pm 0.019$ | $1.715 \pm 0.026$ |
| 16 | 81.818 | $1.626 \pm 0.022$ | $1.676 \pm 0.028$ |
| 17 | 99.709 | $1.529 \pm 0.023$ | $1.506 \pm 0.028$ |
| 18 | 122.071 | $1.695 \pm 0.034$ | $1.767 \pm 0.041$ |
| 19 | 146.253 | $1.698 \pm 0.053$ | $1.768 \pm 0.067$ |
| 20 | 175.287 | $1.449 \pm 0.050$ | $1.381 \pm 0.056$ |
| 21 | 210.057 | $1.579 \pm 0.100$ | $1.721 \pm 0.132$ |
| 22 | 252.322 | $2.589 \pm 0.482$ | $3.419 \pm 1.359$ |
| 23 | 303.256 | $1.876 \pm 0.256$ | $2.285 \pm 0.370$ |
| 24 | 365.513 | $1.495 \pm 0.235$ | $1.579 \pm 0.301$ |

data. Finally, in the late times, the side lobes of boxcar gates align within the frequency of the VLF radios, thus producing strong off-diagonal terms in the covariance plots. Here, the hybrid gates provide better suppression of VLF radios due to lower amplitude side lobes compared to semitapered and box-car gates. Thus, it is clear from the analysis that the Hamming or the semitapered gating scheme is very effective at the late times for the suppression of VLF radios with less distortion in the TEM

signal. This makes it clear that the proposed scheme show overall improvement in terms of time-domain and frequency-domain analysis. Also, gating does not entirely suppress the effect of VLF radios in the resultant gated signals.

The proposed hybrid gating model is configured according to the current towed-TEM system. In the future, data from a fully sampled system will be used by the developed model for the analysis. The data will be gated using fully sampled gates as shown in Fig. 9. For a fair comparison with the existing gating scheme, the proposed work also compares the frequency response of

an existing fully sampled B-spline gate (Nyboe and Mai, 2017; Peng et al., 2022). The frequency response observed for the fully sampled Hamming gate has improved suppression of VLF radios beyond 20 kHz over piece-wise flat Hamming gate. Frequency analysis also reveals that the fully sampled Hamming gate has a better suppression of VLF over existing B-spline gates. It has been observed that Hamming gates have sharp zeros/low magnitudes compared to B-spline gates, offering better possibilities of suppressing some VLF radios completely. Thus, it is clear from the analysis that fully sampled gating with

Hamming gates will have better suppression of VLF over boxcar, semitapered, and piece-wise flat Hamming gates.



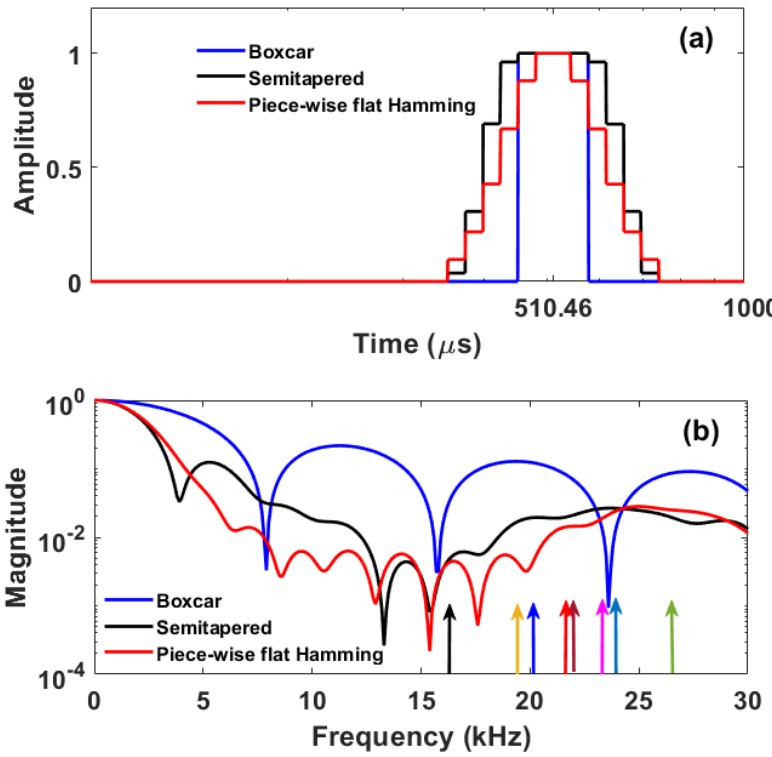

**Figure 8.** Comparative analysis of boxcar, semitapered, and piece-wise flat Hamming gate (Gate number 27) centered at 510.46 $\mu$s. The width of the boxcar gate is 126.75 $\mu$s whereas the width of the semitapered and piece-wise flat Hamming gate is 388.06 $\mu$s. (a) Time-domain analysis and (b) Frequency-domain analysis. The colored arrows in (b) show the frequencies of VLF radios.

## 5.4 Discussion

This section compares the analysis of the current work with existing literature. Larsen et al. explored the usage of semitapered gates to suppress VLF radios (Larsen et al., 2021). The method used a linear combination of boxcar gates to construct a flat-top tapered gate. The semitapered gates are selected such that it minimizes the standard errors with respect to short boxcar gates.
The semitapered gating scheme has obtained an average improvement factor of 1.514. However, optimization of a gate with a single standard error objective may pass some of the VLF radios due to its coherent nature. The analysis only focused on field data, whereas the analysis of synthetic data with different numbers of VLF radios were missing. Also, their analysis lacked covariance between the gates.

Traditionally, the system response for SkyTEM is evaluated using model-based interpolation (Nyboe and Mai, 2017). Nyboe
and Mai used B-spline gating to reduce the data size captured from a fully sampled airborne system (Nyboe and Mai, 2017). The method showed that the high sampling rate and B-spline gating function eliminate the need for a model-based interpolation. However, a detailed discussion about the amount of data and noise reduction achieved using B-spline gating was not given in



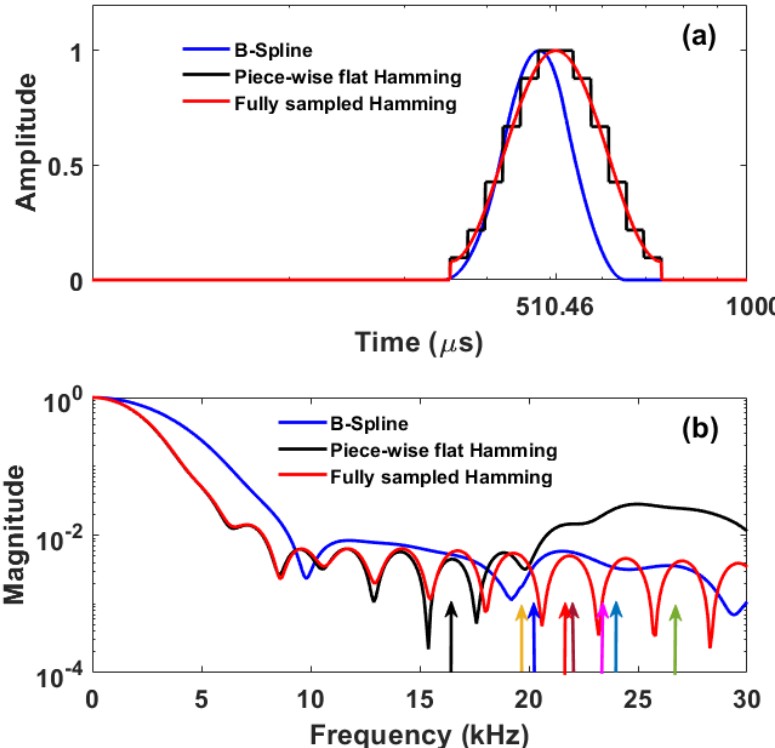

**Figure 9.** Comparative analysis of B-spline (Gate number 27) centered at 472.94 $\mu$s, piece-wise flat Hamming gate, and fully sampled Hamming gates (Gate number 27) centered at 510.46 $\mu$s. The width of the B-spline gate is 308.78 $\mu$s and Hamming gate is 388.06 $\mu$s. (a) Time-domain analysis and (b) Frequency-domain analysis. The colored arrows in (b) show the frequencies of VLF radios.

this paper, nor where qualitative and quantitative model performance for different scenarios using time and frequency domain analysis presented.

Peng et al. demonstrated the usage of B-spline gating for TEM data (Peng et al., 2022). The analysis shows that the suppression of VLF radio at a frequency of 24.1 kHz is better for B-spline gates than short boxcar, long boxcar, and fully tapered gates. The improvement in standard errors of about 15% has been achieved compared to fully tapered gates. However, their analysis does not consider multiple VLF radios active at a single instance of time. Thus, optimizing the frequency response of one B-spline gate for a particular VLF radio, can pass other VLF radios unaltered. Similarly, their analysis did not consider the covariance between the gates, which is common due to the overlapping frequency response of the adjacent gates.

The proposed model overcomes these issues mentioned in the above literature. The proposed model considered three constraints for optimization, namely: minimization of standard errors, covariance, and distortion. The proposed model extends the analysis to synthetic and to field data with varying VLF contents. Analysis shows that the proposed model has obtained collective improvement in standard errors with a minimum standard deviation and suppression in covariance of adjacent gates. The





average improvement factor obtained for the hybrid gating scheme is 1.719 and 1.717 on synthetic and field data, respectively. The analysis also reveals that simulations based on fully sampled gates offer higher suppression in VLF radios than the existing semitapered and B-spline gates. Thus, the proposed model has collectively surpassed the performance of the existing gating scheme for an entire recording duration. The total time required for selecting an optimized gate for one signal was about 17 minutes. Also, the time required to select the optimum gate increased with an increase in the number of iterations and research

agents. It is impractical to use multi-objective particle swarm optimization on a daily basis on, e.g., remote monitoring systems with limited computing resources. Here, an alternative solution is to use the optimization results to generate a set of a few representative and near-optimum gate banks. Deciding which of these gate banks is optimum for any given measurement is a low-computing cost operation.

  The mean of constraints, i.e., the minimum standard error, minimum correlation between off-diagonal terms, and minimum

distortion are used to obtain the optimal combination of gates. The optimal gate is then used to obtain the final gated signal for synthetic and field data. However, in real scenarios, it is not possible to measure the decay of a signal without noise. Therefore, repetition of simulations with only two constraints (minimum standard error, minimum correlation between off-diagonal terms) has also been considered. It has been observed that with two constraints the choice of gate selection varies more frequently from one gate to another (mainly between hybrid and semitapered gates). In addition, the optimum hybrid gate design, is

obtained for a noise model that provides VLF noise of the same amplitude without considering the fluctuations that occurs in real measurements, i.e., daily changes in environmental and atmospheric conditions leading to a varying strength of VLF radio signals. Such variations may affect the optimal gate placement or their types. A different combination of gate shapes may therefore be obtained due to variations in the environmental conditions in field data.

  The CM obtained after gating the field data indicates the presence of a strong pattern at the late times. Thus, the field data

appears to be contaminated with another noise source. A detailed study of daily changes could potentially help identify the noise source(s) and be used in the development of a more robust or adaptive algorithm with improved noise suppression. This, however, is delegated to future investigations.

## 6   Conclusions

  The proposed hybrid gating scheme is a new technique designed for fully sampled and existing TEM systems with analog

boxcar gates. The hybrid gating scheme is a combination of boxcar and Hamming gates. The developed gating technique has provided minimal distortion in the final gated signal, and it also minimizes covariance at mid and late times and standard errors on early and late gates. The developed technique is a promising tool to suppress VLF radio noise during the mid and late times. The covariance analysis provides an analysis of VLF noise and other coherent sources. The developed technique is not only applicable to the existing analog boxcar integrator TEM systems, but will also be ready to be deployed on fully sampled digital

systems. In the future, analysis of more noise sources and solutions to suppress these sources will be examined.



*Data availability.* Data used in this work is available upon request to the corresponding author

*Author contributions.* Smith K. Khare: Data collection, Conceptualization, Methodology, Writing – original draft, and Validation. Paul McLachlan: Data collection, Validation, Reviewing, and Editing. Pradip Kumar Maurya: Data collection and Reviewing. Jakob Juul Larsen: Conceptualization, Validation, Reviewing, and Editing

*Competing interests.* The authors confirm that there is no conflict of interest with this work

*Disclaimer.* Funding Information: This work was supported by Innovation Fund Denmark under grant 0177-00085A.

*Acknowledgements.* The authors would like to thank Innovation Fund Denmark for funding.





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
