# Peer review of "An Optimized and Hybrid Gating Scheme for the Suppression of Very Low Frequency Radios in Transient Electromagnetic Systems"

_Geoscientific Instrumentation, Methods and Data Systems, 2023_

## Referee Comment (RC2)

[referee-annotated manuscript omitted]

---

## Author Response (AR1)

We would like to thank both reviewers for their positive comments. Below are our detailed responses (in italics) and an overview of the changes made to the manuscript.

**RC1.1** You mention the first gate is centered on 0.6μs. How fast do you expect to be able to cut the current? How do you check for any residual influence of the primary field?

*Reply: It is only with the synthetic model, where – by construction - there is no primary field, that we can get reliable results down to 0.6 us. In experimental data, the first gates can indeed be much more troublesome. An example of experimental data with early-time issues is seen in the upper left corner of the four panels in Figure 6 where we find unexpected structures in the covariance matrices, i.e., the covariance matrices aid the quality assurance of data. In this case, we discard early-time data below 10 us as seen in Figure 7. The issue is briefly mentioned in section 5.2.*

**RC1.2** In a monitoring application, did you observe (or do you expect to see) an important variability of the type and shape of the optimised hybrid time gates from one time step to the other? Could a solution to the optimisation efficiency be that the type is determined from an initial extension acquisition and then only the shape is optimized for the following signals?

*Reply: We have not observed any large variations in optimum gate shapes for experimental data. Based on our theoretical analysis we don't expect any large variations either. The common feature of all semi-tapered gates is a significant reduction of amplitude in the frequency response of the sidelobes compared to boxcar gates. Day-to-day variations in noise will change what specific gates are optimum, but the variations between different gate types are small. The approach suggested by the reviewer is indeed a potential solution strategy, but from an implementation efficiency viewpoint we anticipate that the filter bank approach which only needs to compare the output from a few pre-selected gates is the most practical solution.*

**RC1.3** At the end of the discussion, you mention that varying VLF signals could "affect optimal gate placement". However, in the study you considered this parameter as fixed. Do you plan to look into the impact of gate placements on the noise filtering?

*Reply: The short answer is no. First, we would like to emphasize that it is only the amplitudes of the VLF radios that are changing due to atmospheric conditions. The frequencies of VLF radios are highly stable. Second, as seen in Figure 8, the frequency response of a semi-tapered gate is generally smoother and more featureless than the frequency response of a boxcar gate. The important feature is that the sidelobes are overall significantly reduced. Small shifts in gate placement can indeed be expected to increase VLF suppression, but the increase would be very small and hardly worthwhile in practice. The research presented here is intended to improve available boxcar gated data. Long term, the more promising approach is likely to abandon filter-based methodologies and instead investigate model-based subtraction of VLF radio noise (Macnae, 2015).*

**RC1.4**: Line 235: I am not sure if the issue is with the layout, font or something else but I found this sentence, interspersed with equations, very difficult to follow. I would suggest a slight reformulation of the MO-PSO method to increase the readability.

*Reply: The sentence is indeed a bit long. It has been improved in the final version of the manuscript by breaking it into shorter parts.*

**RC2.1** The manuscript is very well written, and the implemented methodology is based on sound principles. I have attached an annotated version of the manuscript where I have highlighted various parts that need attention by the authors. Most of the requested modifications are on typographical mistakes and some unclear statements.

*Reply: The suggested typographical modifications has been made to the final version of the manuscript and the suggestions for cleaning up statements are implemented. In line 267, we have added a comment that the reason distortions in synthetic data start around 100 us is that this is what has been experimentally observed.*

*List of manuscript change in response to reviewer 2.*

*Line 10: "Semitapered" has been changed to "semi-tapered" throughout the manuscript and in figures 4, 7, and 8 as suggested.*

*Line 16. The sentence has been updated to "...utilized in mineral exploration, groundwater mapping, saltwater boundary mapping, and a variety of other applications" in accordance with the reviewer suggestion.*

*Line 23: The sentence has been updated according to reviewer instructions and now reads "as the changes in response caused by temporal groundwater variations are embedded in large background signal.*

*Line 77: "changes in the" has been removed from the sentence as requested.*

*Line 236: The superfluous reference has been removed.*

*Line 238: changed formatting as requested.*

*Line 246: "With" changed to "where" as requested.*

*Line 254: "false" changed to "local" as requested.*

*Line 266: The sentence has been updated from "The values of $A_{RN}$ and $A_{VLF}$ in Eq. (2) are selected such that distortions in resultant TEM signals start after 100 $\mu s$ (Larsen et al., 2021)"*

*To*

*"The values of $A_{RN}$ and $A_{VLF}$ in Eq. (2) are selected such that the visible distortions in resultant TEM signals start after 100 $\mu s$ as observed in TEM data by Larsen et al. (2021)"*

*Line 305 "few" changed to "fewer"*

*Line 370: "Later" has been removed from the sentence to minimize the risk of confusion*

*Line 386: We have removed "by the developed model" to make the sentence more clear*

*Line 398: Layout has been updated*

*Line 399 Text has been updated according to the suggestion*

*Line 410: Layout has been updated*

*Line 424: This is a typo. "Research agents" has been updated to "search agents"*